# The Lipid Phosphate Phosphatase Wunen Promotes Eggshell Formation and Is Essential for Fertility in *Drosophila*

**DOI:** 10.3390/biology12071003

**Published:** 2023-07-14

**Authors:** Amrita Mukherjee, Michaela Schuppe, Andrew D. Renault

**Affiliations:** 1MRC Toxicology Unit, Gleeson Building, Tennis Court Road, Cambridge CB2 1QR, UK; am2634@mrc-tox.cam.ac.uk; 2Institute for Organic Chemistry, Eberhard Karls University Tübingen, Auf der Morgenstelle 18, 72076 Tübingen, Germany; michaela.schuppe@uni-tuebingen.de; 3School of Life Sciences, University of Nottingham, Medical School, QMC, Nottingham NG7 2UH, UK

**Keywords:** follicle cell, egg chamber, oogenesis, chorion, eggshell, *Drosophila*, *wunen*, LPP, fertility

## Abstract

**Simple Summary:**

The eggshell that surrounds insect eggs acts as a barrier, protecting against predators and desiccation. Despite its importance, how this intricate and multilayered structure is produced remains relatively poorly understood. Here, we show that a lipid-modifying enzyme is required by cells of the *Drosophila* ovary for the production of an effective eggshell and fertility, suggesting that lipids are required to regulate these processes. The human version of this enzyme has been implicated in coronary artery disease, and therefore, understanding how this enzyme functions in other organisms may help decipher its role in disease.

**Abstract:**

The eggshell that surrounds insect eggs acts as a barrier, protecting against biotic factors and desiccation. The eggshell is a multi-layered structure which is synthesised by the somatic follicle cells that surround the developing oocyte. Although the temporal order of expression of the protein eggshell components goes someway to explaining how the different layers are built up, but how the precise three-dimensional structure is achieved and how lipid components responsible for desiccation resistance are incorporated are poorly understood. In this paper, we demonstrate that *wunen*, which encodes a lipid phosphate phosphatase, is necessary for fertility in *Drosophila* females. Compared to sibling controls, females null for *wunen* lay fewer eggs which subsequently collapse such that no larvae emerge. We show that this is due to a requirement for *wunen* in the ovarian follicle cells which is needed to produce an ordered and functional eggshell. Knockdown of a septate junction component also results in collapsed eggs, supporting the idea that similar to its role in embryonic tracheal development, Wunen in follicle cells also promotes septate junction function.

## 1. Introduction

The ability of cells to modify their extracellular environment or build extracellular structures is common to both prokaryotes and eukaryotes. Many bacterial species aggregate within biofilms and self-produced matrices of extracellular polymeric substances. Eukaryotic cells produce a wide variety of specialised extracellular structures, ranging from vertebrate bones and insect cuticles to plant cell walls, to name just a few. At the core, this process requires the release of components through secretory pathways. But it may also require cells to organise or build compartments so as to limit the extracellular diffusion of these components.

One example of a complex extracellular structure is the eggshell of insect embryos [1]. Because insects lay eggs that develop externally, they need protection from the environment in the form of an eggshell, or chorion, that protects the egg from desiccation and pathogens, whilst still allowing gaseous exchange.

The *Drosophila* eggshell, like in other insects, is a multi-layered structure. The eggshell is divided into an oocyte proximal vitelline membrane (VM), a lipid wax layer, an inner chorion layer, an endochorion, and a non-proteinaceous exochorion [2]. Eggshell components have distinct temporal expression profiles [3], as reviewed in [4], which partly accounts for how the multi-layered eggshell structure is constructed. Generally speaking, VM proteins are synthesised during the early stages of eggshell formation (stages 8–10) in the form of membrane-bound ‘vitelline bodies’, whereas chorion components are synthesised and secreted later (stages 11–14). However, the timing of synthesis does not always correlate with the final position of the protein in the eggshell, and mutants in particular eggshell components often affect multiple layers (reviewed in [4]). Thus, the different layers cannot be considered simply in isolation, and a complex interplay exists between them.

Both the VM and chorion are stabilised by covalent cross-linking, rendering the eggshell proteins insoluble. For the chorion proteins, this occurs prior to ovulation and involves disulphide and dityrosine cross-linking, involving peroxidase activity within the endochorion. For the VM proteins, disulphide cross-linking occurs before ovulation; however, dityrosine cross-linking only occurs following ovulation (reviewed in [5]). How these cross-linking events are separated spatially and temporally is unclear.

An eggshell is produced during oogenesis by a layer of epithelial follicle cells that surround the developing egg. The follicle cells become specialised for eggshell production in that they undergo three endocycles to become polyploid and then further amplify specific genomic regions, encoding chorion proteins, to enable their high levels of expression [6]. In addition, specialisations of the chorion such as the micropyle, the sperm entry point, and the dorsal appendages, through which gaseous exchange occurs, require subgroups of follicle cells to migrate and modify their behaviour at these specific sites. Thus, the eggshell represents an excellent paradigm to study the many facets needed to manufacture a complex extracellular structure.

Lipid-modifying enzymes play varied and important roles during development and in disease. For example, a class of enzymes called lipid phosphate phosphatases (LPPs) are important in mice for vascular development and for tumour progression via regulation of LPA, an LPP substrate [7]. We have previously described a role for a *Drosophila* LPP called Wunen (Wun) in the tracheal cells of the embryo [8]. These cells must also secrete a number of proteins into the lumen of the trachea to aid modification of the chitin matrix which lines the tracheal lumen [9,10]. Loss of Wun causes a failure of luminal proteins to accumulate, mostly due to their leakage into the surrounding haemolymph due to defective septate junction function [8]. One of the human homologs for Wun, LPP3, has a role in aspects of the early secretory system [11]. Therefore, LPPs may impinge on multiple aspects required to accumulate extracellular proteins, namely promoting intracellular protein trafficking and maintaining extracellular compartment integrity.

In this paper, we have examined the role of Wun during oogenesis based on the observation that *wun* mutant adults are female sterile but male fertile. Here, we show that *wun* mutant females lay relatively few eggs, and these are susceptible to desiccation due to a defective chorion leading to a collapsed egg phenotype and a failure of embryos to hatch into larvae. Wun is expressed and required specifically in the ovarian follicle cells. We show that the amplification of the chorion gene loci is normal in *wun* mutant follicle cells, and these cells can secrete both endogenous and heterologous proteins. However, the integrity of the VM and the chorion structure were greatly perturbed in *wun* mutants, indicating a role for Wun in either coordinating secretion or preventing leakage of secreted material from the follicle cell–oocyte interface. We found that septate junction function is required in ovarian follicle cells to promote eggshell integrity, suggesting possible conservation of Wun function between follicle and tracheal cells.

To explore the biochemical requirement of Wun in follicle cells, we show that Wun function cannot be substituted by Lipin, a phosphatidic acid (PA)-specific phosphatase [12]. We thus conclude that the function of Wun is not the intracellular conversion of PA to diacylglycerol (DAG).

## 2. Materials and Methods

### 2.1. Drosophila Stocks and Handling

*wun^9^* and *wun^23^* are EMS null alleles [13]. The following flies were obtained from the originating labs: *UASwun* [14], *c563 Gal4* [15], *UAS lipin* and *lipin^e00680^* [16], and *Nrg^G00305^* [17]. *UAS cora RNAi* (stock 51845), *Gal80^ts^* (stock 7018), *c355Gal4* (stock 3750), *Cp36^dec2-1^* (stock 4842), *UAS nls.dsRed* (stock 8547), *T155Gal4* (stock 5076), *tubulinGal4* (stock 5138) and *Df(2R)exel7095* (stock 7860) were from the Bloomington stock centre.

For conditional knockdown of *cora*, *c355Gal4/+*;;*Gal80^ts^/UAS cora RNAi* flies were produced and maintained at 18 °C until adults emerged, and these were transferred to 29 °C for 3 to 4 days before being moved to laying cages for egg collection. 

To determine the egg laying and hatching rates, 4 to 5 cages each containing 5 fattened females with equivalent numbers of sibling males were set up, and the flies allowed to lay for 24 h at room temperature. The plates were scored for the number of eggs laid, kept at room temperature for a further 24 h, and scored again for the number of larvae. Each laying cage was scored for 4 consecutive days, and average laying rates were calculated for all cages.

To determine the timing of egg desiccation, eggs were laid at room temperature on apple juice agar collecting plates for 2.5 h, and the proportion of turgid versus collapsed eggs was scored every 1.5 h whilst the plates were kept open at room temperature. The averages are from 4 and 5 repeats for the sibling control and the mutants, respectively.

### 2.2. Immunohistochemistry

The ovaries were dissected in PBS and fixed in 4% formaldehyde for 20 min. The ovaries were washed in PBS with 0.1% Tween (PBST) and then blocked in PBST with 0.1% BSA. The following primary antibodies were used: goat anti-Nudel at 1:50 (dL-20, Santa Cruz Biotechnology, Santa Cruz, CA, USA), mouse anti-GFP at 1:350 (JL-8, Clontech, Mountain View, CA, USA), chicken anti-GFP at 1:1000 (13970, Abcam, Cambridge, UK), mouse anti-BrdU at 1:20 (Becton Dickinson, Franklin Lakes, NJ, USA), goat anti-Nudel at 1:100 (dc-16, Santa Cruz Biotechnology), and mouse anti-Crumbs at 1:50 (Cq4, DSHB, Iowa City, IA, USA). Secondary antibodies were used at 1:500 (Cy3 conjugates, Jackson ImmunoResearch, Philadelphia, PA, USA) or 1:1000 (Alexa488 conjugates, Invitrogen, Waltham, MA, USA). TRITC-Phalloidin at 1:500 (Invitrogen) was used to label the actin. The ovaries were mounted in Aquamount (PolySciences, Warrington, PA, USA) and imaged using an Olympus FV-1000 or Leica SP2 confocal microscope. BrdU labeling was carried out as described in Calvi and Lilly (Methods in Molecular Biology).

Detection of extracellular ANF-GFP was carried out based on [18]. The ovaries were dissected in a drop of cold Schneider’s media, teased apart with a sharp tungsten needle, transferred to a microcentrifuge tube, and washed with ice-cold PBS. The ovaries were incubated for an hour with a mouse anti-GFP antibody in PBS with 0.1% BSA. The ovaries were washed three times with ice-cold PBS and then fixed using 4% formaldehyde in PBS for 20 min on ice. The ovaries were washed twice with PBST and then stained as described above.

Eggshell opacity was analysed by viewing the embryos mounted in PBS. The neutral red assay was carried out as described in [19] using 5 mg/mL neutral red (Sigma, St. Louis, MO, USA) in PBS.

### 2.3. Electron Microscopy

For transmission electron microscopy of the *Drosophila* ovaries, freshly dissected ovaries were cryo-fixed by high-pressure freezing by transferal to aluminium platelets of 150 or 200 μm depth containing 1-hexadecene. The platelets were sandwiched with platelets without any cavity and then frozen with a high-pressure freezer (Bal-Tec HPM 010, Balzers, Balzers, Liechtenstein). The frozen samples were freed from extraneous hexadecene under liquid nitrogen and transferred to 2 mL microtubes with screw caps (Sarstedt #72.694) containing 1 ml of the frozen substitution medium and 2% OsO4 and 0.5% uranyl acetate (the uranyl acetate was prepared from a 20% stock solution in methanol) in anhydrous acetone. After transferring the samples into a Leica AFS-2 freeze substitution unit, freeze substitution was started by warming up the samples to −90 °C followed by a freeze substitution protocol of −90 °C—52 h, −90 °C to −60 °C—3 h, −60 °C—6 h, −60 °C to −40 °C—3 h, and −40 °C—6 h. At the end of the −60 °C step, glutaraldehyde was added from an aqueous 25% solution (EMS #16220, Electron Microscopy Sciences, Ft. Washington, DC, USA) to give a 0.5% glutaraldehyde and 2% water concentration. After the end of the −40 °C step, the samples were placed on ice for 1 h and then washed 3 times with cold acetone. Infiltration was started with 20% resin in ethanol on ice, followed by 50% resin, 80% resin each for 1 h, 100% resin overnight, and 2 × 100% resin each for 1 h at room temperature on a rotating wheel. For the resin, we used an Araldite 502/Embed 812 kit (EMS #RT 13940, Electron Microscopy Sciences) mixed according to the manufacturer’s instructions. Finally, the ovaries were placed in flat embedding molds and cured at 60 °C for 48 h. Ultrathin sections (50–70 nm) stained with 2% uranyl acetate in H20 for 10 min and in 0.4% lead citrate in 0.1 N NaOH for 2 min were examined with a FEI Tecnai G2 BioTwin transmission electron microscope operated at 120 kV. Images were recorded with a Gatan USC 4000 digital camera in the full resolution of 4 k × 4 k.

For the scanning electron microscopy of the eggs, the eggs were fixed in 70% acetone at 4 °C overnight, dehydrated through an ethanol series, subjected to critical point drying from CO_2_, followed by sputter coating with 10 nm Au-Pd in a Balzers MED 010 sputter coater (Balzers, Liechtenstein). The samples were examined at 20 kV accelerating voltage in a Hitachi S800 field emission scanning electron microscope.

## 3. Results

### 3.1. Wunen Is Required during Oogenesis in Egg Chamber Follicle Cells

To investigate the cause of the sterility of the *wun* null females, we examined the egg laying rates and egg quality from *wun* transheterozygous females. Such females lay more than 80% fewer eggs than their sibling controls (Figure 1A). In addition, these eggs fail to hatch into larvae (Figure 1A).

To examine where in the ovary *wun* is required, we performed in situ hybridisation using a *wun* probe on adult ovaries. We found that *wun* is strongly expressed in the follicle cells of egg chambers from stage 10 onwards (Figure 1B). In germ cell migration during embryogenesis, Wun acts redundantly with its paralogue Wun2 with which it shares a similar expression pattern during early embryogenesis [20]. We therefore examined whether *wun2* was also expressed in follicle cells. Whilst we observe *wun2* expression in the nurse cells at stage 10, we did not see follicle cell expression (Figure 1C). This suggests that the sterility caused by the loss of just *wun* is because *wun2* is not expressed in follicle cells. A similar situation occurs in tracheal development in the embryo in which tracheal defects are observed in *wun* mutants alone but can be rescued by either *wun* or *wun2* ectopic expression [8].

To test whether the egg laying defects of *wun* adult females result from the loss of follicle cell expression, we attempted to rescue the egg laying and hatching defects by expressing Wun using two different follicle cell drivers, *T155 Gal4* [21] and *c563 Gal4* [15]. The former drives expression in follicle cells from stage 10 onwards, whilst the latter does so only from stage 13, as judged using a *UAS ns.dsred* reporter (Figure 1D,E). *wun* transheterozygous females carrying *T155 Gal4* and *UAS wun* showed increased egg laying rates compared to *wun* transheterozygotes, although not to wild type levels (Figure 1A). In addition, many of the eggs laid hatched into larvae, indicating that follicle cell expression was sufficient to restore fertility to otherwise *wun* null females. When using the *c563* driver, however, only a slight increase in egg laying was observed, and almost no eggs hatched (Figure 1A).

Taken together, we conclude that *wun* is required in somatic cells of the ovary, namely the follicle cells for egg laying and fertility. Furthermore, Wun is required before stage 13, which is before follicle cells begin the secretion of eggshell components.

### 3.2. Lipin, an Intracellular PA Phosphatase, Cannot Substitute for Wun in Follicle Cells

Wunens are LPPs (formerly termed type 2 phosphatidic acid phosphatases: PAP2), integral membrane proteins that generally show broad substrate specificity in vitro (reviewed in [22]. Wun, for example, can dephosphorylate a number of phosphorylated lipid substrates in vitro including PA, lysophosphatidic acid, and ceramide 1-phosphate [23].

To probe the biochemical nature of the activity provided by Wun in follicle cells, we wanted to test whether a more substrate-specific lipid phosphatase could rescue *wun* loss of function in follicle cells. One such enzyme family are the Lipins, which are evolutionarily unrelated to LPPs but show lipid phosphatase activity (formerly termed type 1 phosphatidic acid phosphatases: PAP1) and a high degree of specificity towards PA [24].

Expression of Lipin in a *wun* mutant background caused a slight increase in egg laying, but this was much lower than the rescue with *wun* (Figure 1A). In addition, the hatching rates were not rescued (Figure 1A). We conclude that Lipin is not able to substitute for Wun in follicle cells. We also tested the reciprocal situation and asked whether Wunens are able to substitute for the loss of Lipin. *lipin* transheterozygous null mutants are adult lethal [16] (Table 1). This lethality can be rescued by the expression of Lipin from a ubiquitous *tubulin Gal4* driver (Table 1). However, the expression of Wun or Wun2 using this driver was unable to rescue lethality (Table 1). We conclude that Wun and Lipin, in general, have non-overlapping functions and that the function of follicle cell Wun is not the intracellular conversion of PA to DAG.

### 3.3. Eggs Laid by Wun Mutant Females Are Susceptible to Desiccation

We examined the eggs laid by *wun* mutant females to determine why they did not hatch. The eggs were often collapsed with dorsal appendages that were flaccid and intertwined (Figure 2A,B). The number of dorsal appendages was wild type, indicating that there was no patterning, in particular ventralisation, defects. To ascertain if the eggs were laid in such a fashion or if they desiccated after being laid, we scored the percentage of eggs appearing collapsed over time following a narrow laying time-window.

The eggs laid by the sibling control females were turgid and remained so for the following 24 h, after which they hatched into larvae (Figure 2C). The eggs laid by *wun* transheterozygous mothers were also initially nearly all turgid, but over time, the proportion that collapsed gradually increased, such that by 24 h, more than 50% were no longer turgid (Figure 2C). We conclude that the collapsed eggs are not caused by a failure to be properly filled during oogenesis but rather a susceptibility to desiccation once laid.

### 3.4. Eggs Laid by Wun Mutant Females Exhibit Eggshell and VM Defects

Eggs are protected from desiccation by their eggshell. To determine if and how the eggshell is affected in eggs from *wun* mutant females, we firstly examined it by scanning EM. The surface of the chorions laid by the sibling control females shows pentagonal/hexagonal elevated ridges that correspond to the cell borders of the follicle cells that secreted the eggshell [2]. Inside the polygons, the surface is uneven but in a stereotyped fashion (Figure 3A) due to the pillar structures that make up the endochorion [2]. In contrast, the majority of chorions in eggs laid by the *wun* null females completely lacked ridges (five of the eight eggs examined compared to none of the nine eggs from the sibling controls), and the surface was not uneven in the stereotyped fashion of the wild type (Figure 3B). Instead, we observed small deep holes on the chorion surface (eight of the eight eggs examined), which we never observed in the wild type (none of the nine eggs from the sibling controls).

In other cases, follicle cell impressions were present, but were barely visible (three of the eight eggs examined (Appendix A). Both the ridges and the uneven surface of the chorion were restored in eggs laid by *wun* mutant females with *wun* expressed in follicle cells (six of the six eggs examined) (Figure 3C).

We next examined the opacity of the chorion both as a measure of its thickness coupled to its resistance to extraction in water by examining the eggs under transmitted light mounted in PBS. The eggs from the sibling control females were opaque, and the enclosed embryo was not visible through the chorion (Figure 3D). In contrast, the eggs from the *wun* transheterozygous females were translucent, with the encapsulated egg being properly visible through the chorion (Figure 3E). In the eggs laid by the rescue females the opacity of the chorion was restored, however not to the extent in the wild type, with regions of opacity especially away from the ridges (Figure 3F). Taken together these data demonstrate that Wun is required for the generation of an opaque insoluble chorion.

We also examined the integrity of the VM which lies between the embryo and the chorion and acts as a barrier to small molecules, including the dye neutral red. We tested whether the eggs from the control and mutant females were resistant to staining by neutral red. Very few of the eggs from the control females were stained by neutral red, indicative of an effective VM (Figure 3G). However, 22% of the eggs from the *wun* mutant females became stained (Figure 3H). This percentage is comparable to published data, and our own analysis of a mutant in a gene encoding the endochorion protein *Cp36* in which 30% of eggs were stained (Figure 3J,K) [19].

The VM defects were rescued by the expression of *wun* in the follicle cells (Figure 3I). We conclude that Wun is required in follicle cells for a functional VM.

### 3.5. Egg Chamber Organisation and Follicle Cell Functionality Are Not Compromised by the Loss of Wun

To determine the cause of the eggshell defects caused by the loss of *wun*, we examined the egg chambers in the *wun* null females. Firstly, we examined the overall egg chamber morphology, in particular the arrangement of the follicle cells. Although several abnormal egg chambers were visible, for the most part, the egg chambers appeared to be wild-type, with border cells migrating to the oocyte and a continuous layer of follicle cells surrounding the oocyte (Figure 4A,C).

Genomic loci containing genes encoding chorion proteins are specifically amplified during the late stages of oogenesis (stage 10B onwards) through repeated rounds of DNA replication of these specific loci [6]. Loss of this amplification, such as in *chiffon* mutants, results in the production of eggs with thin chorions without ridges [25]. Amplification sites can be identified as subnuclear foci that strongly incorporate BrdU [6]. We used this test to examine whether amplification was occurring correctly in the *wun* mutant follicle cells. In the wild type, one to two puncta of BrdU incorporation per follicle cell nucleus were observed, as has previously been reported (Figure 4B). This pattern was identical in the *wun* mutants (Figure 4D). We conclude that chorion gene amplification occurs normally in *wun* mutants.

To examine whether follicle cell polarisation or apical morphology were affected by the loss of *wun,* we examined the localisation of a Cadherin, Cad99C, which specifically localises to follicle cell apical microvilli [26]. In both the sibling control and the *wun* transheterozygous female follicle cells, Cad99C localised to the apical surface in a spiky pattern consistent with localisation to microvilli (Figure 4E–H). We conclude that Wun is not required for correct follicle cell polarisation or microvilli.

We next examined whether the follicle cells of *wun* mutants were able to secrete both endogenous and heterologous proteins. Nudel is a serine protease that is secreted by follicle cells into the perivitelline space and is important for dorsal ventral patterning as well as chorion protein cross-linking [27,28]. In both the sibling control and the *wun* mutant egg chambers, Nudel accumulated around the oocyte (Figure 4I,J). To examine the ability of the follicle cells to secrete a heterologous protein, we expressed a fluorescently tagged version of rat atrial natriuretic factor, ANF-GFP [29], in the follicle cells. We specifically detected secreted GFP by performing anti-GFP antibody incubation prior to fixation and solubilisation. In the sibling control and the *wun* mutant females, secreted GFP was visible surrounding the oocyte (Figure 4K,L), although it was not as uniformly distributed as compared to Nudel (Figure 4I,J). This is likely due to differing levels of expression of the transgene in the follicle cells, as evidenced by the non-uniform intracellular endogenous GFP fluorescence (the green channel in Figure 4K,L), with high levels of secreted GFP being detected adjacent to follicle cells with high levels of intracellular GFP. Taken together, these data demonstrate that *wun* mutant follicle cells retain their bulk secretory ability.

### 3.6. Eggshell Organisation Is Disrupted by the Loss of Wun

To examine the chorion structure of the oocytes in the *wun* mutants in more detail, stage 14 egg chambers in the *wun* mutants were examined by TEM (Figure 4M–P). The sibling control egg chambers showed the stereotyped multi-layered chorion (Figure 4M,O). The oocyte is surrounded by the VM followed by a very thin inner chorion layer. Surrounding this is the endochorion with a thin fenestrated floor (inner endochorion) separated from a thicker roof (outer endochorion) by vertical pillars generating cavities. The roof is not flat but instead many thin processes (the roof network) extend into the exochorion. The latter usually appears as two layers, with the innermost layer being less electron-dense than the outer [2].

In the *wun* mutants, although the basic multi-layered structure is intact, the endochorion is much thinner than in the control, is not continuous, and lacks the thin processes that normally extend into the exochorion (Figure 4N). The exochorion is also much less electron dense, and the two layers are not discernible. In contrast, the VM is present and of the same thickness as the controls (Figure 4O), and the inner chorion layer is indistinguishable from the controls (Figure 4N). We conclude that *wun* is required for the accumulation of exo- and endochorion material.

### 3.7. Wun Localszes to Follicle Cell Plasma Membranes

To ascertain where in follicle cells Wun may be functioning, we examined the localisation of a GFP-tagged Wun construct. WunGFP is functional as evidenced by its ability to rescue *wun* induced defects in the *Drosophila* embryonic tracheal system as well as its ability to disrupt germ cell migration similar to untagged Wun [8]. We found that WunGFP localised predominantly to the lateral surfaces of the follicle cell plasma membrane but with some accumulations of intracellular GFP towards the apical side (Figure 5A). WunGFP localises to the plasma membrane in other tissues including embryonic mesodermal cells [23]. In polarised cells, however, such as tracheal epithelial cells, WunGFP does not localise along the entire lateral surface but is restricted to an apicolateral region, overlapping with septate junction components [8].

Follicle cells are also polarised epithelial cells and also express septate junction components. Follicle cell septate junction components localise along the entire lateral membrane until stage 11 [30,31,32] and only become apicolateral restricted late in oogenesis [33]. Incipient SJs between follicle cells form as early as in stage 6 egg chambers, but typical pleated SJs are only observed from stage 10 onwards [34,35], as reviewed in [36].

To compare the localisation of septate junctions and WunGFP we examined the localisation of a GFP protein trap in the septate junction component Neuroglian (NrgGFP). In agreement with previous findings, we see NrgGFP localising along the entire lateral follicle cell membrane (Figure 5B). We conclude that the lateral localisation of WunGFP, but not the apical intracellular accumulations, resembles other septate junctions’ components, indicating a possible role of Wun in septate junction function.

### 3.8. Septate Junction Knockdown in Follicle Cells Leads to Reduced Egg Laying and Collapsed Eggs

To test if the loss of septate junction function could phenocopy the *wun* mutants, we wanted to knock down the expression of the essential septate junction *coracle* (*cora*) in the follicle cells [37]. To test the knockdown efficiency, we expressed a *cora* RNAi construct in embryonic tracheal cells (Figure 6B), a cell type that requires septate junctions for paracellular barrier function and gas filling at the end of embryogenesis [38]. *cora* knockdown in these cells prevented tracheal gas filling in 100% of embryos (Figure 6A–C), which phenocopies *cora* mutants (our unpublished results). We conclude that this RNAi line is able to cause penetrant *cora* knockdown.

Knockdown of *cora* using the follicle cell driver *c355Gal4* produced no adults, suggesting that some earlier knockdown of *cora* expression was leading to lethality. We therefore incorporated a temperature-sensitive Gal80, Gal80^ts^, to induce knockdown specifically in adults. The flies were reared to adults at the permissive temperature and then either maintained at this temperature or placed at a restrictive temperature to initiate *cora* knockdown. At the permissive temperature, the resulting females were fertile, laying wild-type eggs (Figure 6D,E) which hatched. At the restrictive temperature, the resulting females were sterile and laid only a very small number of eggs (too few to be reliably scored), which were collapsed (Figure 6F), and none hatched, rendering the females sterile. We conclude that follicle cell septate junctions are essential for egg production and integrity.

## 4. Discussion

In this paper, we have characterised the reduced fecundity and fertility exhibited by *wun* mutant females. We have shown that the eggs laid by *wun* transheterozygote females have defects in several layers of their eggshell which leave them susceptible to desiccation. Through tissue-specific rescue experiments, we have shown that Wun is required specifically in the ovarian follicle cells that surround the developing egg. This requirement seems specific to the somatic cells rather than the germline as evidenced by previous analysis of *wun* mutant germ line clones which produce wild-type-looking fertile eggs which are not susceptible to desiccation [39].

Wun is required in follicle cells between stages 10 and 13 based on the endogenous *wun* expression pattern and the rescue experiments with the Gal4 drivers that switch on at different stages of oogenesis (Figure 1). Stages 10 to 13 correspond to those during which the follicle cells synthesise the VM and chorion proteins. We have shown that in *wun* mutants, the amplification of chorion genes occurs as in the wild type and that the follicle cells appear morphologically normal and are able to secrete proteins into the oocyte–follicle cell space, suggesting that *wun* is not affecting these cellular processes.

This leaves several hypotheses that would be able to explain the eggshell phenotype. The first possibility is that in *wun* mutants, the secretion of selected VM and chorion proteins is affected or that the ordering of their secretion is disrupted. In support of the latter, in mutants, such as *ptx*, that disrupt the timing of chorion gene expression, eggshell morphogenesis is affected which also leads to VM permeability defects [40]. *wun* mutants might have this phenotype because of altered cargo trafficking within follicle cells.

The second possibility is that although these proteins are secreted correctly, they fail to accumulate to sufficient levels in the space between the follicle cell and oocyte due to leakage into the haemolymph. SJ components are expressed in ovarian follicle cells, and these would be predicted to provide such a barrier between the oocyte and the haemolymph. This function would be identical to the role of *wun* in the embryonic tracheal epithelial cells where it is required for SJ-dependent accumulation of Serp and Verm in the tracheal lumen [8].

The knockdown of the SJ component *cora* in follicle cells resulted in both reduced egg laying and collapsed eggs (Figure 6), phenotypes both seen for *wun,* indicating possible involvement for *wun* in follicle cell SJ function. However, both phenotypes were much more severe than for the *wun* mutants: The number of eggs after *cora* knockdown was so low as to be unscorable, and they either desiccated immediately or were already collapsed upon laying.

Theoretically, this might result from a partial redundancy between *wun* with *wun2*, as is observed in germ cell migration [39], but we do not see *wun2* expression in follicle cells (Figure 1). This suggests that any role for *wun* in promoting follicle cell SJ function is not absolute, unlike in tracheal cells where the loss of *wun* causes a complete loss of Serp and Verm from the lumen [8].

Although septate junctions’ components are expressed in follicle cells (Figure 5) [30,31,32], it is not known if they provide a barrier that prevents the diffusion of components between the oocyte–follicle cell space and the haemolymph. By bathing freshly dissected ovaries in 10 and 70 kDa fluorescent dextran, we were unable to demonstrate a barrier in the stage 10 egg chambers (our unpublished results), so either there is no complete barrier that surrounds the oocyte at this stage, despite the presence of follicle cell septate junctions [34,35], or it is highly sensitive to dissection.

A further open question is the extent to which the rate of egg laying and embryo survival into larvae are coupled. In particular, *wun* expression using the *c563* Gal4 partially rescues egg laying but not survival from *wun* null mothers (Figure 1A). It may be that the low number of eggs is due to a requirement for *wun* elsewhere in the ovary/adult. Perhaps both follicle cell Gal4 drivers express elsewhere in the ovary/adult, which is sufficient to partially increase egg laying, but only the *T155* Gal4 drives Wun expression early enough in the follicle cells to rescue the eggshell sufficiently to allow the embryos to hatch into larvae. 

This work adds to the growing list of morphogenetic processes in which LPPs are required in *Drosophila*, including tracheal and heart development and germ cell migration. There is no known role for SJ in germ cell migration, suggesting that LPPs have both SJ and SJ-independent effects. The ability of some *wun* homologues to rescue *wun* mutant phenotypes in some tissues but not others is suggestive of different biochemical activities being required in the different tissues. Finding lipid metabolic genes whose mutations phenocopy *wun* will be useful in narrowing down the specific lipids involved in each case.

## 5. Conclusions

We conclude that Wunen is required autonomously in ovarian follicle cells of the *Drosophila* ovary for the production of a functioning eggshell. Wunen function cannot be substituted by another lipid-dephosphorylating enzyme, in this case Lipin, indicating that the critical Wunen substrate is not shared by Lipin (and vice-versa) or that the subcellular localisation of these enzymes is different and critical to their function. 

## Figures and Tables

**Figure 1 biology-12-01003-f001:**
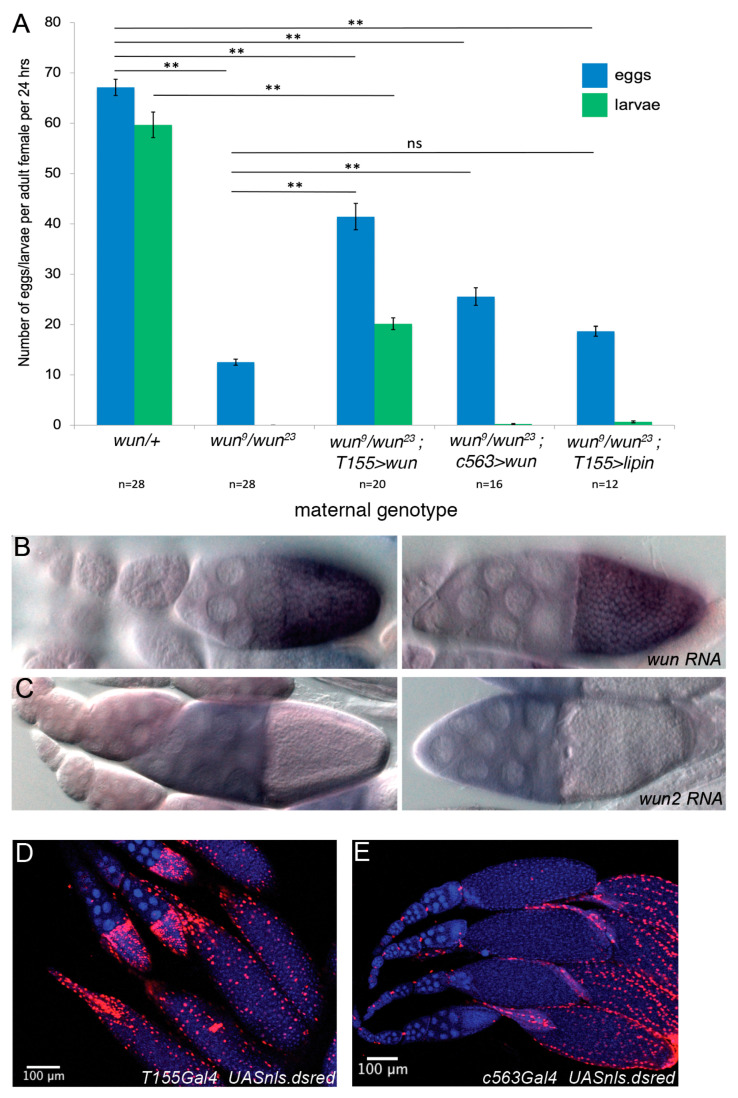
Wunen is expressed and required in follicle cells for oogenesis. (**A**) A graph showing the average number of eggs laid and subsequent larvae per adult female per 24 h. The error bars show the standard error of the mean. ** *p* < 0.01 (one-way ANOVA with Tukey’s HSD test); ns: no significant. n = number of 24 h layings scored. (**B**,**C**) *wun* (**B**) and *wun2* (**C**) in situ hybridisation pattern showing expression in the follicle cells of the stage 9 (left) and stage 10A egg chambers (right). (**D**,**E**) Expression pattern of Gal4 driver lines *T155* (**D**) and *c563* (**E**) as demonstrated by crossing to *UAS nls.dsred* flies and staining the ovaries of resulting females with DAPI (blue) to mark nuclei with endogenous DsRed fluorescence (red).

**Figure 2 biology-12-01003-f002:**
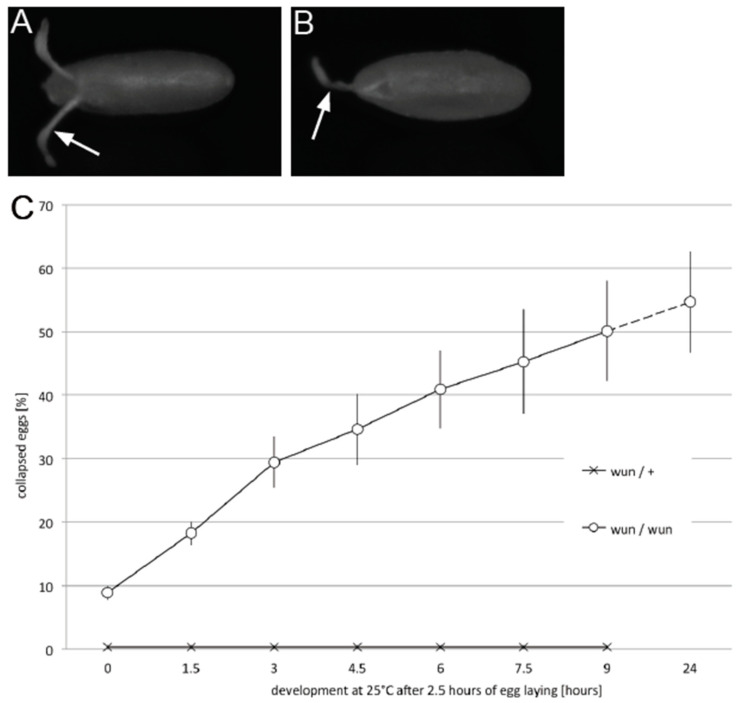
Eggs laid by *wun* mutant females desiccate with time. (**A**,**B**) Egg laid by the *wun/+* sibling control (**A**) and the *wun^9^/wun^23^* (**B**) females showing dorsal appendages (arrows), which in the mutants are flaccid and often intertwined. (**C**) Graph showing average percentage of collapsed (desiccated) eggs over time following 2.5 h egg laying by the *wun/+* sibling control and the *wun^9^/wun^23^* females on an apple juice agar plate. The averages are from 4 and 5 layings with a total of 450 and 253 eggs scored, respectively. The error bars show the standard error of the mean. Dashed line indicates discontinuity in the *x*-axis scale.

**Figure 3 biology-12-01003-f003:**
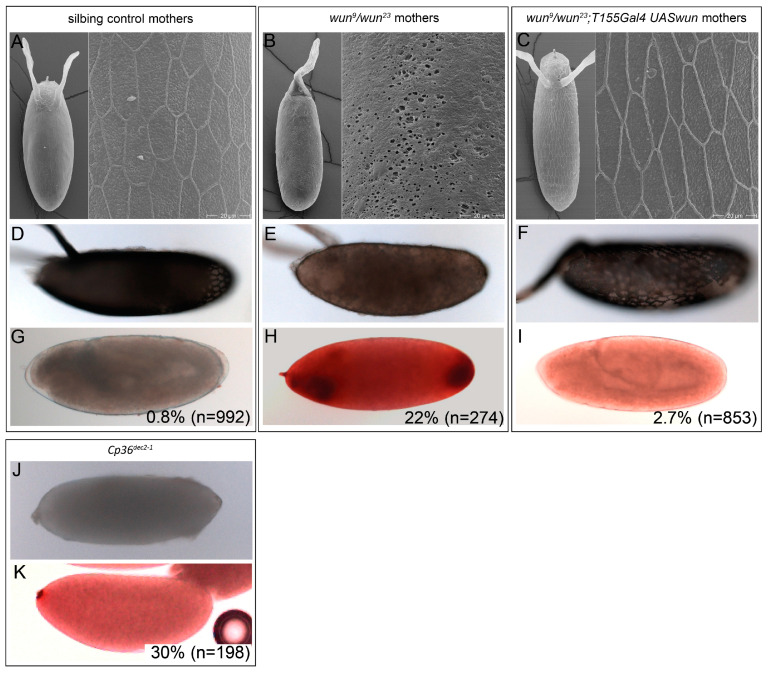
Eggs laid by *wun* mutant females have defective chorions and VMs. |The eggs laid by the sibling control (**A**,**D**,**G**), *wun^9^/wun^23^* (**B**,**E**,**H**), and *wun^9^/wun^23^*;*T155Gal4/UASwun* (**C**,**F**,**I**), and *Cp36^dec2-1^* mothers (**J**,**K**). (**A**–**C**) Scanning EM micrographs of the whole eggs (left) and the magnified portion of the eggshell surface (right). (**D**–**F**,**J**) Transmitted light images of the embryos mounted in PBS showing eggshell opacity. (**G**–**I**,**K**) Transmitted light images of embryos following incubation with neutral red to indicate the permeability of the VM. The numbers indicate the percentage of eggs positive for neutral red staining and the total number of eggs scored.

**Figure 4 biology-12-01003-f004:**
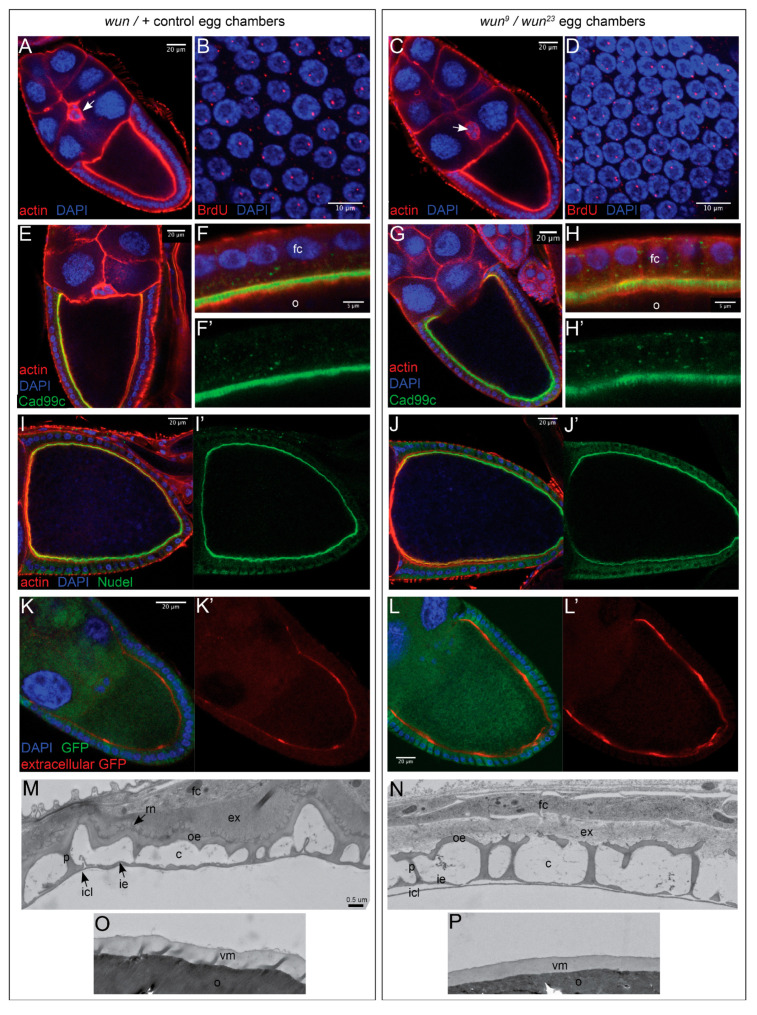
*wun* mutant follicle cells amplify chorion genomic loci and show normal secretion of endogenous and heterologous proteins but fail to organise a proper eggshell. Egg chambers from sibling control (**A**,**B**,**E**,**F**,**I**,**K**,**M**,**O**) and *wun^9^/wun^23^* (**C**,**D**,**G**,**H**,**J**,**L**,**N**,**P**) females. (**A**,**C**) The mutant stage 10 egg chambers show a correctly organised follicle cell layer and the migration of border cells (arrows), as judged using phalloidin to mark actin (red) and DAPI to mark nuclei (blue). (**B**,**D**) Follicle cells exposed to BrdU and then fixed and stained with DAPI (blue) and an anti-BrdU antibody (red) to highlight sites of DNA amplification, which at this stage correspond to loci containing clusters of genes encoding eggshell proteins. (**E**–**H**) Stage 10 egg chambers stained for DAPI (blue), phalloidin (red), and an antibody against the apically localised follicle cell protein Cadherin99C (green). (**F**,**H**) Higher magnification images of the interface between the follicle cells (fc) and oocyte (o). (**I**,**J**) Stage 10 egg chambers stained using DAPI (blue), phalloidin (red), and an antibody against the secreted protein Nudel (green). (**K**,**L**) Follicle cells expressing the secretion reporter ANF-GFP using a *T155* Gal4 driver using DAPI to mark nuclei (blue), an anti-GFP antibody used prior to permeabilisation to mark only extracellular GFP (red), and endogenous GFP fluorescence (green). The intensity differences in the extracellular GFP staining are due to the non-uniform expression of the ANF-GFP reporter in the follicle cells. (**M**–**P**) TEM micrographs of an egg chamber at the interface between a follicle cell (fc) and oocyte (o) showing the chorion structure, including the exochorion (ex), the roof network (rn), the outer endochorion (oe), pillars (p), the cavity of the endochorion (c), the inner endochorion (ie), the inner chorionic layer (icl), and the vitelline membrane (vm). vm is from the same egg chamber and of the same magnification, but it is shown as a separate panel because the vm has separated from the chorion as a result of fixation. (**F’**,**H’**,**I’**,**J’**,**K’**,**L’**) show the single channel image for Cad99c (**F’**,**H’**), Nudel (**I’**,**J’**) and extracellular GFP (**K’**,**L’**) from their respective multichannel panels.

**Figure 5 biology-12-01003-f005:**
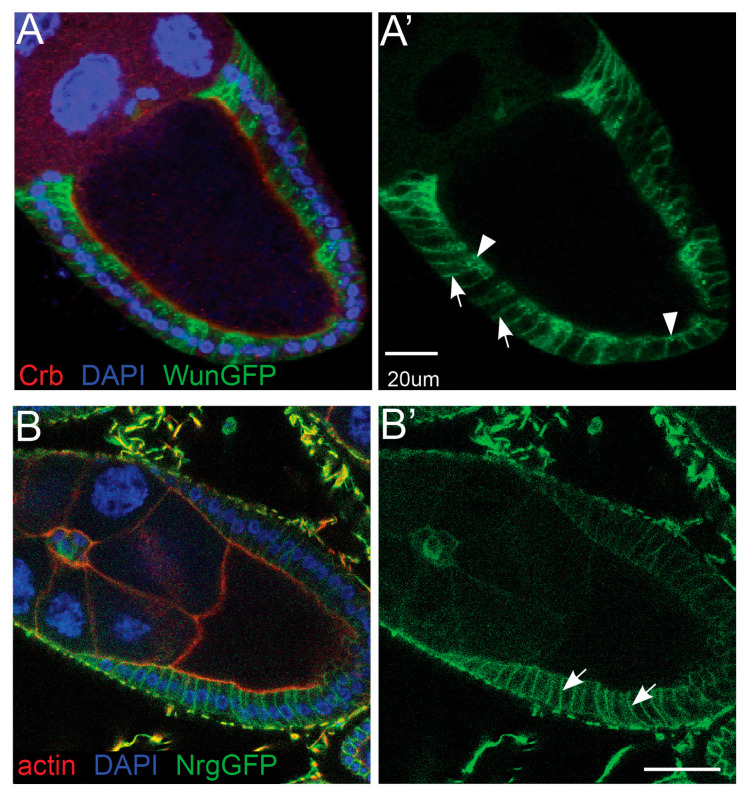
Wunen localises to the plasma membrane. (**A**) Stage 9–10 egg chamber expressing WunGFP in follicle cells under the control of the *T155* Gal4 driver. Arrows indicate lateral plasma membrane localisation, and arrowheads indicate apical intracellular accumulations. (**B**) NrgGFP gene trap line showing the location of the SJs along the entire lateral domains of the follicle cells. The egg chambers were stained using an anti-GFP antibody (green) and co-stained with an anti-Crumbs antibody to mark the apical surfaces (red, (**A**)) or phalloidin to show actin (red, (**B**)) and DAPI to mark nuclei (blue, (**A**,**B**)). The arrows indicate lateral plasma membrane localisation. (**A’**,**B’**), show the single channel image for WunGFP (**A’**) and NrgGFP (**B’**) from their respective multichannel panels.

**Figure 6 biology-12-01003-f006:**
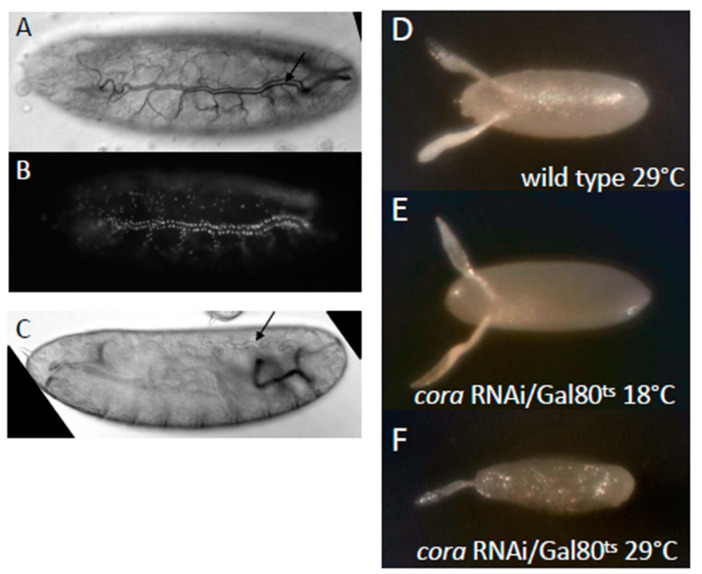
Knockdown of *coracle* in follicle cells causes collapsed eggs and defective eggshells. (**A**–**C**) Lateral views of stage 16 embryos laid by *btlGal4* females crossed to *UAS nls.dsred* (**A**,**B**) or *UAS cora* RNAi males showing wild-type gas filling of the trachea ((**A**), arrow) and florescent expression of *nls.dsred* in the tracheal cells (**B**) or a lack of gas filling ((**C**), arrow) upon *coracle* knockdown. Eggs from the wild-type (*white*) females kept at 29 °C (**D**), *c355Gal4/+*;;*Gal80^ts^/UAS cora* RNAi females (**E**,**F**) kept at 18 °C (**E**) and 29 °C (**F**) for 3–4 days prior to laying.

**Table 1 biology-12-01003-t001:** Table showing the viability of *lipin* transheterozygous adults when *lipin*, *wun,* and *wun2* are expressed using a *tubulin Gal4* driver.

Genotype	Viability
*lipin^e00680^/Df(2R)exel7095; tub Gal4/+*	lethal
*lipin^e00680^/Df(2R)exel7095; tub Gal4/UAS lipin*	viable and fertile
*lipin^e00680^/Df(2R)exel7095; tub Gal4/UAS wun*	lethal
*lipin* *^e00680^/Df(2R)exel7095; tub Gal4/UAS wun2myc*	lethal

## Data Availability

Raw data presented in this study are available on request from the corresponding author.

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
