# Peer review of "The Lipid Phosphate Phosphatase Wunen Promotes Eggshell Formation and Is Essential for Fertility in Drosophila"

_biology, 2023, doi:10.3390/biology12071003_

Round 1
Reviewer 1 Report
This is a well-presented manuscript that clearly defines a role for Wunen in eggshell formation.
Apart from minor issues my only major comment is that the authors show that Wunen is expressed in follicle cells, localises to the plasma membrane in a similar manner to septate junction components and then go on to show that loss of septate junctions results in a similar phenotype to wun mutants. They also state that in trachea Wunen is required for correct localisation of septate junction components – why then not also show that this is the case in follicle cells as it would complete the story?
Minor issues:
Line 105: should be “UAS cora-RNAi” and not “UAS cora”.
Figure 3J: I am unsure why this is included. If it is to demonstrate a lack of eggshell opacity should the embryo not have a chorion – it appears to be dechorionated.
Figure 4B, D – the DAPI staining is difficult to see. Could the contrast be increased? Also, are the nuclei smaller in panel D and is this significant?
Line 355: This sentence structure appears to be incorrect.
Author Response
1) why then not also show that this is the case in follicle cells as it would complete the story?
SJ components were originally reported to be localised along the entire lateral membrane of Drosophila ovarian follicle cells (eg. Nrg in Wei, Hortsch and Goode 2004 Dev Dynamics). It was only more recently that enrichment at the apical side was reported (eg. Cora in Alhadyian, Shoaib, and Ward 2021 G3) which only occurs late in oogenesis when the follicle cells are very flat. When we carried out these experiments we did not observe enrichment at late stages however our experiments were done before the Alhadyian et al. paper.
Minor issues:
2) Line 105: should be “UAS cora-RNAi” and not “UAS cora”.
Corrected
3) Figure 3J: I am unsure why this is included. If it is to demonstrate a lack of eggshell opacity should the embryo not have a chorion – it appears to be dechorionated.
Mutant or RNAi mothers lay eggs with dysmorphic dorsal appendages and the chorion is very thin (Cernilogar et al. 2001 Dev Genes Evol;Velentzas et al. 2016 Sci Reports), hence why it looks dechorionated already. It is shown for completeness of the figure.
4) Figure 4B, D – the DAPI staining is difficult to see. Could the contrast be increased? Also, are the nuclei smaller in panel D and is this significant?
On the screen the DAPI staining looks comparable in brightness. There might be some issue when printed as blue can be harder to see. The nuclei density is slightly different which is most likely due to the slight difference in the stage of the oocyte. There were no obvious difference in nuclei size observed in the mutant compared to wild type.
5) Line 355: This sentence structure appears to be incorrect.
This has been changed.
Reviewer 2 Report
Wunen, a lipid phosphate phosphatase, has been shown to be involved in various morphogenetic processes including development and cell migration. The work done by Mukherjee et al. shows a novel role for wun in reproduction and more specifically its role in egg maintenance. Wun mutants exhibited lower rates of egg laying in comparison to wild type controls and upon investigation these eggs were improperly structured and susceptible to desiccation. Expression patterns of wun were examined and visualized to be restrictive to the follicle cells in egg chamber development. This was confirmed with their rescue experiments that targeted specific stage follicle cell expression and showed that wun mutant egg defects could be rescued with wun expression in follicle cells specifically in stages 10-13. The authors provided several hypotheses that could provide an explanation for the defects seen with the wun mutants, however their leading hypothesis was that defective septate junctions caused leakage of chorion and VM proteins that may be contributing to susceptibility to desiccation and improper formation. To test this they targeted a known septate junction component, cora, and found similar phenotypes to that of the wun mutant in decreased egg laying and desiccation susceptibility. These phenotypes, however, were more severe than seen in the wun mutants, although not much data is shown. Overall, the study is well executed and a few suggestions are recommended below to improve the manuscript.
Major issues:
1) In Figure 1A, the authors show that c563-GAL4 partially rescues egg laying but not larval survival. The authors should provide an explanation for how the egg laying and larval survival could be unconnected.
2) The images in Figure 3 are convincing for the eggshell defect. From the images shown in Fig 2B and 3B, it appears as if the dorsal appendages are fused. Is there any evidence of a ventralization defect? Along these lines, the authors should show later stages of oogenesis (stages 11-14) in figure 4 to document that there are not any patterning defects in addition to the eggshell defect.
3) The data presented showing that cora is required for eggshell production are not particularly convincing – only one egg is shown at low magnification. Some analysis similar to the work shown for wunen should be done such as examining the ovaries for normal morphology and a specific defect in eggshell production, or this statement should be revised “We conclude that follicle cell septate junctions are essential for egg production and proper eggshell formation.”
Minor issues:
1) Given the redundancy with wun2 in embryogenesis, it would be interesting if the authors could mention if expression of wun2 in follicle cells could rescue the eggshell defect.
2) Given wunen’s role in germ cell death in the embryo, it would be interesting for the authors to show if nurse cell death is altered in the wunen mutants.
Author Response
1) In Figure 1A, the authors show that c563-GAL4 partially rescues egg laying but not larval survival. The authors should provide an explanation for how the egg laying and larval survival could be unconnected.
It may be that the low number of eggs is due to a requirement of wunen elsewhere in the ovary/adult. Perhaps both Gal4 drivers express elsewhere in the ovary/adult which is sufficient to partially increase egg laying, but only the T155 Gal4 drives Wun expression early enough in the follicle cells to rescue the egg shells sufficiently to allow the embryos to hatch into larvae.
This text has been added to the discussion (line 480).
2) From the images shown in Fig 2B and 3B, it appears as if the dorsal appendages are fused. Is there any evidence of a ventralization defect? Along these lines, the authors should show later stages of oogenesis (stages 11-14) in figure 4 to document that there are not any patterning defects in addition to the eggshell defect.
Although it might appear that the dorsal appendages are fused but they are not - they are just intertwined and can be teased apart. So there is no evidence of a ventralization defect. A sentence to such effect has been added to the results (line 237). We would have to do additional stainings to specifically test for subtle patterning defects, but the unfused nature of the dorsal appendages leads us to believe this is not necessary.
3) The data presented showing that cora is required for eggshell production are not particularly convincing – only one egg is shown at low magnification. Some analysis similar to the work shown for wunen should be done such as examining the ovaries for normal morphology and a specific defect in eggshell production, or this statement should be revised “We conclude that follicle cell septate junctions are essential for egg production and proper eggshell formation.”
Whilst we would like to be able to expand the analysis as per the reviewers suggestion, two hurdles arise. Firstly, I have moved institutions since the initial EMs were done on Wunen, therefore it would be tricky to repeat this for cora. Secondly, the number of cora eggs laid is very small. A cage of 60 females only lay a handful of eggs in an overnight collection. Therefore we have amended the text on line 420 to change "and eggshell formation" to "and egg integrity".
4) Given the redundancy with wun2 in embryogenesis, it would be interesting if the authors could mention if expression of wun2 in follicle cells could rescue the eggshell defect.
We have tried that experiment. Several of our UAS constructs (including UAS wun2) only result in very patchy expression in follicle cells (high in some cells and low in others). This may be due to the genomic insertion site. Therefore, we did not include this data.
5) Given wunen’s role in germ cell death in the embryo, it would be interesting for the authors to show if nurse cell death is altered in the wunen mutants.
While we did not specifically look for cell death of nurse cells, we have not observed dying nurse cells in wun mutants. Wun results in germ cell death upon over-expression in somatic cells. We have over-expressed wun in follicle cells and we have not observed cell death in the follicle cells or nurse cells. This is similar to our previous work where over-expressing wun in tracheal cells does not cause cell death (Ile et al. 2012 Development). Its role in cell death upon over-expression may be specific to germ cells and only when they are migrating.
Round 2
Reviewer 2 Report
The authors have addressed my concerns.